# Bootstrapped SSL CycleGAN for Asymmetric Domain Transfer

Lidija Krstanović [1,*] , Branislav Popović [1] , Marko Janev [2] and Branko Brkljač [1]

1 Faculty of Technical Sciences, University of Novi Sad, Trg D. Obradovića 6, 21000 Novi Sad, Serbia; bpopovic@uns.ac.rs (B.P.); brkljacb@uns.ac.rs (B.B.)
2 Institute of Mathematics, Serbian Academy of Sciences and Arts, Kneza Mihaila 36, 11000 Belgrade, Serbia; markojan@uns.ac.rs
* Correspondence: lidijakrstanovic@uns.ac.rs

**Abstract:** Most CycleGAN domain transfer architectures require a large amount of data belonging to domains on which the domain transfer task is to be applied. Nevertheless, in many real-world applications one of the domains is reduced, i.e., scarce. This means that it has much less training data available in comparison to the other domain, which is fully observable. In order to tackle the problem of using CycleGAN framework in such unfavorable application scenarios, we propose and invoke a novel Bootstrapped SSL CycleGAN architecture (BTS-SSL), where the mentioned problem is overcome using two strategies. Firstly, by using a relatively small percentage of available labelled training data from the reduced or scarce domain and a Semi-Supervised Learning (SSL) approach, we prevent overfitting of the discriminator belonging to the reduced domain, which would otherwise occur during initial training iterations due to the small amount of available training data in the scarce domain. Secondly, after initial learning guided by the described SSL strategy, additional bootstrapping (BTS) of the reduced data domain is performed by inserting artifically generated training examples into the training poll of the data discriminator belonging to the scarce domain. Bootstrapped samples are generated by the already trained neural network that performs transferring from the fully observable to the scarce domain. The described procedure is periodically repeated during the training process several times and results in significantly improved performance of the final model in comparison to the original unsupervised CycleGAN approach. The same also holds in comparison to the solutions that are exclusively based either on the described SSL, or on the bootstrapping strategy, i.e., when these are applied separately. Moreover, in the considered scarce scenarios it also shows competitive results in comparison to the fully supervised solution based on the pix2pix method. In that sense, it is directly applicable to many domain transfer tasks that are relying on the CycleGAN architecture.

**Keywords:** CycleGAN architecture; semi-supervised learning; bootstrapping; imbalanced data

## 1. Introduction

The task of image-to-image domain translation (or much broader domain transfer task) is to transfer or translate images (i.e., to learn the mapping) from one domain into another, while preserving the content, where the term domain denotes a particular style (if images are for example drawings), season (like winter or summer, if images are photographs), etc. This can be achieved by using sets of paired training samples (images), which is referred as the supervised domain adaption, or by using unpaired training samples, which is also known as the unsupervised domain adaption. Domain adaption refers to a similar task.

Transfer of images from one source domain into another target domain found its application in various image processing, computer graphics, computer vision and related problems. For example, in semantic image synthesis [1–4], image-to-image translation [5–8], image inpainting [9,10], image super-resolution [11,12], etc.

In the case of supervised approaches that are mostly based on the conditional Generative Adversarial Networks (GANs, or cGANs), and which were developed first, all

images are assumed to be available in pairs corresponding to the samples denoting the same thing in the source and the target domain (i.e., in the "first" and the "second" domain). Isola et al. [13] have first applied cGAN in the domain translation task, by proposing supervised pix2pix learning strategy. This approach was further developed by [14–16], but those solutions still failed to capture the complex structural relationships of the scenes in the cases when, e.g., two domains have drastically different views, i.e., while trying to achieve mapping through a single translation network.

Nevertheless, despite the efforts to improve the supervised methods, for many domain transfer tasks paired training images are not fully available. Therefore, besides the described approaches, almost at the same time an unsupervised domain transfer method known as CycleGAN was proposed in [17]. It utilizes two GANs oriented in opposite directions, i.e., from one domain into another, and vice versa. As introduced mappings that are learned are highly under-constrained, cycle consistency loss was introduced in [17] to reduce the mentioned anomaly.

Cycle consistency constraint proved to be very effective as a technique for preserving semantic information of the data with respect to a task of domain transfer, and is therefore applied in various tasks, e.g., in image-to-image translation [17], emotion style transfer [18], speech enhancement [19], etc. However, many of these problems, such as emotion style transfer, speech enhancement, speaker domain transfer, medical image domain transfer, have inherent domain asymmetry, in the sense that one of the domains has significantly less training data available. Therefore, such domain is usually named as reduced or *scarce* in comparison to another one, which is considered as fully observable. According to the authors' knowledge, only several approaches tackled the mentioned problem in the literature, i.e., analyzed domain transfer in cases when one of the two domains is scarce. In [20], an augmented cyclic adversarial learning model that enforces the cycle-consistency constraint via an external task-specific model is proposed. Additionally, in [21], the authors add semi-supervised task-specific attention modules to generate images that are used for improving the performance in a medical image classification task. However, in all those papers, a task-specific approach is used in forming additional cost functions, which means that the analysis of each specific task is required for problem formulation, as well during the training procedure.

In this work, we propose a different approach. We combine the original unsupervised CycleGAN architecture with a semi-supervised learning (SSL) strategy, as well as with a bootstrapping (BTS) procedure. SSL is used to prevent overfitting of the CycleGAN discriminator that is belonging (or being related) to the assumed scarce domain in the considered domain transfer mapping. Discriminator overfitting could occur during the initial learning iterations of the unsupervised CycleGAN due to insufficient number of training samples in the scarce domain, and consequently lead to an underperforming GAN. On the other hand, the second part of the proposed method, i.e., the bootstrapping strategy (BTS), has the aim to bootstrapp the statistics of the problematic discriminator belonging to the scarce domain (i.e., its training pool statistics). This is accomplished by inserting (bootstrapping) artificially generated example data into the discriminator's training pool from time to time during the training procedure. Namely, the corresponding generator residing in the second, fully observable domain of the CycleGAN is periodically called during the training to randomly produce the necessary samples in the scarce domain, and thus compensate the fact that the original training pool of the discriminator in the scarce domain was inadequate. This sampling procedure is applied periodically during the training, but for the first time only after the necessary generator is already sufficiently well trained, i.e., after the initial SSL based on a small amount of paired samples from both domains is performed. Elements of the described BTS-SSL method, i.e., combined SSL and BTS procedure ("SSL+BTS") that is applied during the proposed SSL of CycleGAN in the considered asymmetric domain size scenarios, are illustrated in Figure 1.

The initial SSL strategy that is based on a small set of paired observations allows discriminator as well the generator that are residing in the scarce domain to avoid overfitting

and learn necessary parameters to some extent. In the latter stages of the training procedure the learning process is further guided by the combined SSL as well as by the usual adversarial training strategy, but with an additional augmentation of the discriminator's training pool in the scarce domain, from time to time. This is illustrated in the right part of Figure 1, where it is highlighted that the training involves three types of data, as compared to only one in the baseline unsupervised case. Actually, during the training, after a number of initial iterations, the pool corresponding to the discriminator of the scarce domain is constantly (periodically) filed by training examples generated by the network *F* in Figure 1, which is transferring the training examples from the full to the scarce domain.

Thus, contrary to the approach proposed in [20], where the domain attention mechanism is utilized, the proposed BTS-SSL model is independent of the specific domain transfer task, and requires only a relatively small amount of paired examples in the very beginning of the training procedure. This property makes it directly applicable to different tasks.

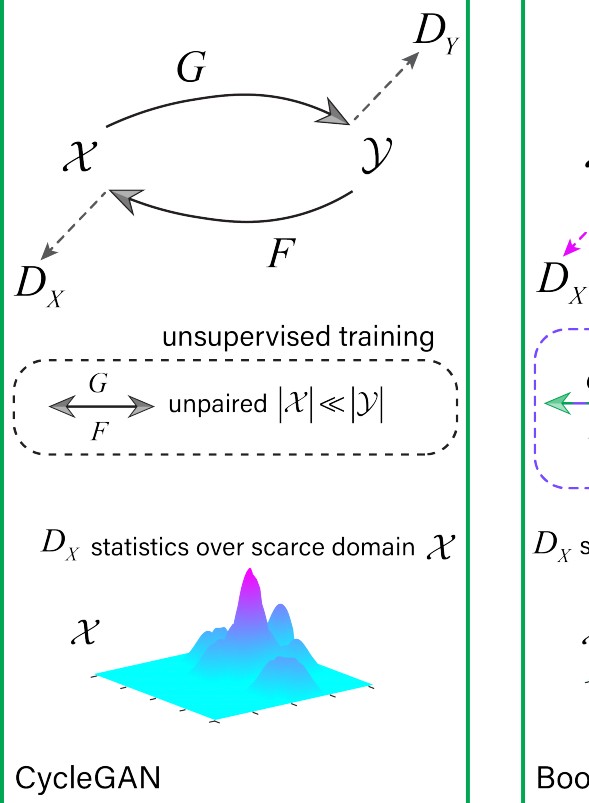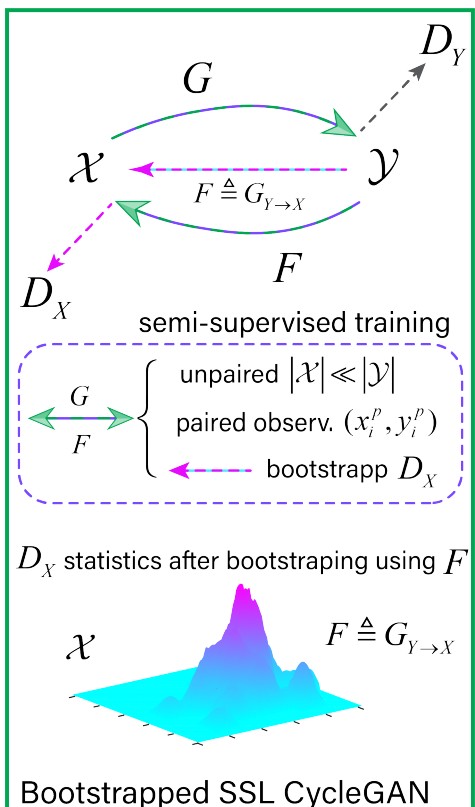

**Figure 1.** Elements of the proposed SSL method, as compared to the original CycleGAN architecture. It is assumed that one of the domains, e.g., $\mathcal{X}$ is scarce, i.e., $|\mathcal{X}| \ll |\mathcal{Y}|$, which makes the unsupervised CycleGAN training much harder. Therefore, the proposed BTS-SSL CycleGAN is invoked.

The paper is organized as follows. Firstly, in Section 2 we give the preliminaries on GANs as well as CycleGAN training strategies, on which we further build the proposed method. Secondly, in Section 3 we introduce the combined SSL and a novel bootstrapping strategy (BTS) in the context of CycleGAN architecture, in order to deal with imbalanced domain transfer tasks that are characterized by one of the two domains being scarce. In Section 4 we describe the elements of the adopted network architectures, training details and experimental setup. Finally, in Section 5 results of extensive experiments involving different training strategies, domain transfer tasks, and varying level of scarcity are presented and discussed. At the end, based on gathered insights, final conclusions are drawn in Section 6.

## 2. Preliminaries

In this section, we give an overview of methods and algorithms that we relate to and further adapt in the paper.

### 2.1. Generative Adversarial Networks

In [22], the ground-breaking method of learning the true data distribution in a non-parametric way was proposed in the form of Generative Adversarial Networks (GANs). In that constellation, the discriminator network $D$ is trying to discriminate between the synthetic samples artificially generated by the generator network $G$ and the ground truth observations available in the training data. The generator $G$ models the true data distribution by learning to confuse, i.e., deceive the discriminator $D$ by providing $D$ with synthetic examples that are hardly distinguishable from the real ones. Thus, the two structures, the discriminator $D$ and the generator $G$, are mutually competing in order to reach the Nash equilibrium expressed by the minimax loss of the training procedure, where the optimization problem is given by:

$$\min_{G} \max_{D} \quad \mathbb{E}_{x \sim p(x)} \left[ \ln \left( D(x) \right) \right] + \mathbb{E}_{z \sim p(z)} \left[ \ln \left( 1 - D \left( G(z) \right) \right) \right], \tag{1}$$

where $p(x)$ represents the true data distribution over domain $\mathcal{X}$, while the latent variable $z$ is sampled by the distribution $p(z)$.

The first term in Equation (1), $\ln \left( D(x) \right)$, corresponds to the cross-entropy between true distribution $[\, 1 \;\; 0\,]^T$ and distribution $[\, D(x) \;\; 1 - D(x)\,]^T$ corresponding to discriminator's probabilistic output. Similarly, the second term: $\ln \left( 1 - D(G(z)) \right)$, is the cross-entropy between $[\, 1 \;\; 0\,]^T$ and $[\, D(G(z)) \;\; 1 - D(G(z))\,]^T$.

In Figure 1, generators in both directions are assumed to be in form of GANs, and are denoted by mappings $F$ and $G$, while the corresponding discriminators over domains $\mathcal{X}$ and $\mathcal{Y}$ are denoted by $D_X$ and $D_Y$.

### 2.2. CycleGAN Networks

The CycleGAN Network is designed to capture special characteristics of one image collection and to figure out how these could be translated into another image collection in the absence of any supervisor, i.e., paired training examples, as finding those is difficult, as well as expensive in the sense of labelling effort that has to be employed simultaneously over two different domains.

In [17], Zhu et al. have proposed invoking the "*cycle consistency*" loss in the overall loss function, by designing two domain translators or mappers $G$ and $F$, in the mutually opposite domain-wise directions. Namely, by learning:

$$G : \mathcal{X} \to \mathcal{Y}, \tag{2}$$

$$F : \mathcal{Y} \to \mathcal{X}, \tag{3}$$

in the form of GAN networks, and by encouraging both mappings given by Equations (2) and (3) and illustrated in Figure 1, to be as "close" as possible to bijection, i.e., by making:

$$F \left( G(x) \right) \approx x, \tag{4}$$

$$G \left( F(y) \right) \approx y, \tag{5}$$

it was shown to be possible to compensate the lack of paired data samples, and effectively learn the complex nonlinear transformations in a fully unsupervised manner.

Therefore, if we denote the generator networks implementing mappings $G$ and $F$ by $G_{X \to Y}$ and $G_{Y \to X}$, respectively, and denote the discriminator networks corresponding to

domains $\mathcal{X}$ and $\mathcal{Y}$, i.e., true distributions $p_X(x)$ and $p_Y(y)$, by $D_X$ and $D_Y$, respectively, from Equation (1) we get the following adversarial optimization objectives:

$$\mathcal{L}_{adv}(G_{X\to Y}, D_Y) = \mathbb{E}_{y\sim p_Y(y)}\big[\ln D_Y(y)\big] + \mathbb{E}_{x\sim p_X(x)}\big[\ln(1 - D_Y(G_{X\to Y}(x)))\big] \quad (6)$$

$$\mathcal{L}_{adv}(G_{Y\to X}, D_X) = \mathbb{E}_{x\sim p_X(x)}\big[\ln D_X(x)\big] + \mathbb{E}_{y\sim p_Y(y)}\big[\ln(1 - D_X(G_{Y\to X}(y)))\big] \quad (7)$$

as well as the cycle-consistency objective:

$$\mathcal{L}_{cyc}(G_{X\to Y}, G_{Y\to X}) = \mathbb{E}_{x\sim p_X(x)}\big[\,\|G_{Y\to X}(G_{X\to Y}(x)) - x\|\,\big] \quad (8)$$

$$+ \mathbb{E}_{y\sim p_Y(y)}\big[\,\|G_{X\to Y}(G_{Y\to X}(y)) - y\|\,\big] \quad (9)$$

making the full optimization objective given by:

$$\mathcal{L}(G_{X\to Y}, G_{Y\to X}, D_X, D_Y) = \mathcal{L}_{adv}(G_{X\to Y}, D_Y) + \mathcal{L}_{adv}(G_{Y\to X}, D_X)$$

$$+ \lambda_{cyc}\,\mathcal{L}_{cyc}(G_{X\to Y}, G_{Y\to X}) \quad (10)$$

the one being optimized during the unsupervised training of the model shown in the left part of Figure 1. Effect of the enforced cycle-consistency during the training (learning) is regulated by the penalty parameter $\lambda_{cyc}$.

### 2.3. Unsupervised Training Procedure

In the following lines of Algorithm 1 are formally summarized the main steps of the unsupervised training procedure that was discussed in more details in Section 2.2. The algorithm implements an iterative optimization of the objective function given by Equation (10).

The presented steps are the basis for the proposed semi-supervised extension of the original CycleGAN algorithm, presented in Section 3.

---

**Algorithm 1** CycleGAN training procedure

---

**procedure** CYCLEGAN (process in the left part of Figure 1)

  $N$, number of iterations; $m$, minibatch size; $\eta > 0$, learning rate; $X \in \mathcal{X}, Y \in \mathcal{Y}$, unpaired or unlabeld training sets, such that $|X| \ll |Y|$;

  Randomly initialize parameters of discriminators $D_X$, $D_Y$, and generators $G_{X\to Y}$, $G_{Y\to X}$:

  $\theta_{D_X}, \theta_{D_Y}, \theta_{G_{X\to Y}}, \theta_{G_{Y\to X}}$

  **for** $k = 1$ to $N$ **do**

   Sample minibatch of unpaired training data $\{x_1, \ldots, x_m\} \subset X, \{y_1, \ldots, y_m\} \subset Y$

   $\widehat{\mathcal{L}}_{adv}(G_{X\to Y}, D_Y) = \frac{1}{m}\sum_{i=1}^{m} \ln D_Y(y_i) + \frac{1}{m}\sum_{i=1}^{m} \ln\left(1 - D_Y(G_{X\to Y}(x_i))\right)$

   $\widehat{\mathcal{L}}_{adv}(G_{Y\to X}, D_X) = \frac{1}{m}\sum_{i=1}^{m} \ln D_X(x_i) + \frac{1}{m}\sum_{i=1}^{m} \ln\left(1 - D_X(G_{Y\to X}(y_i))\right)$

   $\theta_{D_X}^{(k+1)} \longleftarrow \theta_{D_X}^{(k)} - \eta\,\nabla_{\theta_{D_X}}\widehat{\mathcal{L}}_{adv}(G_{Y\to X}, D_X)$

   $\theta_{D_Y}^{(k+1)} \longleftarrow \theta_{D_Y}^{(k)} - \eta\,\nabla_{\theta_{D_Y}}\widehat{\mathcal{L}}_{adv}(G_{X\to Y}, D_Y)$

   $\theta_{G_{Y\to X}}^{(k+1)} \longleftarrow \theta_{G_{Y\to X}}^{(k)} - \eta\,\nabla_{\theta_{G_{Y\to X}}}\left[\widehat{\mathcal{L}}_{adv}(G_{Y\to X}, D_X) + \lambda_{cyc}\,\mathcal{L}_{cyc}(G_{X\to Y}, G_{Y\to X})\right]$

   $\theta_{G_{X\to Y}}^{(k+1)} \longleftarrow \theta_{G_{X\to Y}}^{(k)} - \eta\,\nabla_{\theta_{G_{X\to Y}}}\left[\widehat{\mathcal{L}}_{adv}(G_{X\to Y}, D_Y) + \lambda_{cyc}\,\mathcal{L}_{cyc}(G_{X\to Y}, G_{Y\to X})\right]$

  **end for**

**end procedure**

---

## 3. Proposed Approach

In order to deal with the mutually imbalanced domains, i.e., a problem where one of the domains involved in the translation task is scarce, we utilize an approach with two additional concepts, which are added to the standard CycleGAN model. Nevertheless, as in the original CycleGAN architecture, the proposed solution remains task-independent and avoids introduction of any task-dependent optimization constraints. The general idea of the method is depicted by the functional diagram in the right part of Figure 1.

### 3.1. Avoiding the Overfitting by SSL Strategy

Firstly, we employ the semi supervised learning (SSL) strategy based on the simple $\|\cdot\|_1$ norm error term, which is computed over all pairs or elements in the provided small set of labeled or paired training samples $(x_i^p, y_i^p)$ that represent the same thing in the involved scarce $\mathcal{X}$ and a fully observable domain $\mathcal{Y}$, respectively.

Thus, despite an unfavourable situation in which the number of unlabeled training samples in domain $\mathcal{X}$ is considered as significantly lower than the number of available unlabeled observations in the domain $\mathcal{Y}$, i.e., $|\mathcal{X}| \ll |\mathcal{Y}|$, it is assumed that there still exists a small amount of labeled training pairs $(x_i^p, y_i^p)$ that are possible to be reliably matched into pairs based on their ground-truth labels, and thus provide the basis for the SSL strategy.

Based on the valuable information contained in this small set of $M_{SSL}$ paired observations, i.e., $\mathcal{P}_{data} = \{(x_i^p, y_i^p)|i = 1, \ldots, M_{SSL}\}$, it is possible to define an additional SSL optimization term $\mathcal{L}_{SSL}$, which will enforce that the learned CycleGAN mappings in Equations (2) and (3) respect the fact that $x_i^p \in \mathcal{X}$ and $y_i^p \in \mathcal{Y}$ are paired, i.e., have the same label. This is illustrated in Figure 2, where the expected closeness of the samples $x_p$ and $y_p$ after transformation of $x_p$ by the learned mapping $G$ is depicted.

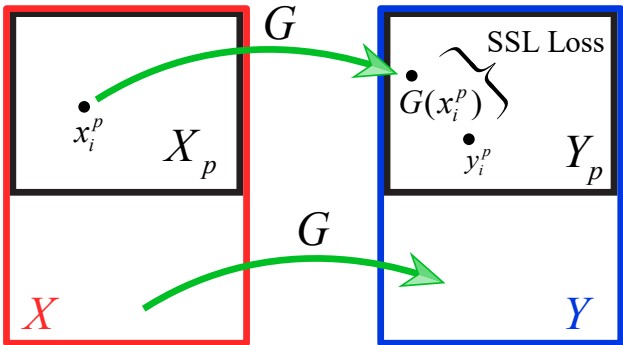

**Figure 2.** SSL loss $\mathcal{L}_{SSL}$, Equation (11), which enforces closeness of the "transformed" $\hat{y}_i^p = G(x_i^p)$ and the "true" $y_i^p$ over a small set of paired, i.e., labeled samples $\mathcal{P}_{data} = \{(x_i^p, y_i^p)|i = 1, \ldots, M_{SSL}\}$ from a scarce domain $\mathcal{X}$ and a fully observable $\mathcal{Y}$. The same also holds in the opposite direction, for $F(y_i^p)$.

As the result of previous considerations, the closeness of the "transformed" $G(x_i^p)$ and the "true" labeled samples $y_i^p$ during the SSL phase of the proposed training procedure can be described by the following optimization objective:

$$\mathcal{L}_{SSL}(G_{X \to Y}, G_{Y \to X}) = \frac{1}{m} \sum_{i=1}^{m} \left[ \|G_{X \to Y}(x_i^p) - y_i^p\|_1 + \|G_{Y \to X}(y_i^p) - x_i^p\|_1 \right], \quad (11)$$

which evaluates and imposes the corresponding closeness over a predefined amount of $M_{SSL}$ available paired examples in both directions, from $\mathcal{X}$ into $\mathcal{Y}$, as well to the opposite side, from $\mathcal{Y}$ into $\mathcal{X}$, Figures 1 and 2.

Thus, by applying the described SSL procedure and the introduced $\mathcal{L}_{SSL}$ objective, we prevent discriminator $D_X$ in the scarce domain $\mathcal{X}$ to overfit during the initial iterations of the learning process. In the standard unsupervised learning setting, characteristic for any CycleGAN model, overfitting of discriminator $D_X$ would mostly be inevitable, due

to assumed limited number of available unlabeled training samples in the scarce domain, i.e., adopted assumption that $|\mathcal{X}| \ll |\mathcal{Y}|$.

By combining Equation (11) with the standard CycleGAN objective presented in Equation (10), the full SSL objective is made of:

$$
\begin{aligned}
\mathcal{L}(G_{X \to Y}, G_{Y \to X}, D_X, D_Y) = \ & \mathcal{L}_{adv}(G_{X \to Y}, D_Y) + \mathcal{L}_{adv}(G_{Y \to X}, D_X) \\
& + \lambda_{cyc}\, \mathcal{L}_{cyc}(G_{X \to Y}, G_{Y \to X}) \\
& + \lambda_{SSL}\, \mathcal{L}_{SSL}(G_{X \to Y}, G_{Y \to X}),
\end{aligned}
\tag{12}
$$

where $\lambda_{SSL}$ is the corresponding regularization parameter.

### 3.2. Improving the Learning Process by BTS Strategy

As the second part of the proposed learning procedure, we employ the bootstrapping (BTS) strategy based on the internal structures already available in the CycleGAN model.

Since the domain $\mathcal{X}$ is assumed to be scarce, during the training discriminator $D_X$ at its disposal has a significantly smaller pool of unlabeled training samples in comparison to the discriminator $D_Y$ in the domain $\mathcal{Y}$. Therefore, the aim of the proposed BTS strategy is to overcome this imbalance, by artificially expanding the size of the unlabeled training pool of the discriminator $D_X$. This is illustrated in the left part of Figure 3.

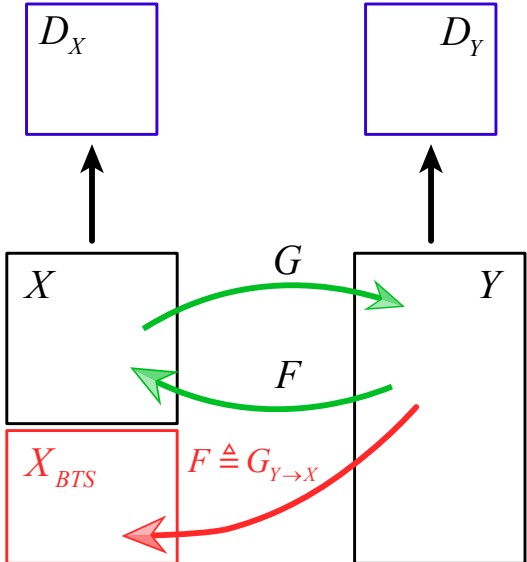

**Figure 3.** Bootstrapping of the statistics of the discriminator $D_X$ in the scarce domain $\mathcal{X}$, by adding a set of novel examples $X_{BTS}$ to the original pool of unlabeled training data $X$. Bootstrapped samples are randomly generated by $G_{Y \to X}$.

This is achieved by exploiting the already learned mapping $F$, shown in the right part of Figure 1, i.e., the GAN implementing the generator function $G_{Y \to X}$. Under the term "already learned", it is assumed that in the initial phase of the training procedure, parameters of the CycleGAN model are first initially optimized for some time using the previously described SSL strategy from Section 3.1. Only after this "initial training" phase, the $G_{Y \to X}$ structure from the CycleGAN model should be considered as reliable enough to be used as the bootstrapping data sampler for the training pool of the discriminator $D_X$ residing in the scarce domain $\mathcal{X}$. Of course, the decribed BTS strategy could also be applied independently from the SSL one, but in the assumed scarce $\mathcal{X}$ scenarios such approach would not result in the same level of performance, as in the "SSL+BTS" case, when these two strategies are synergistically combined together. The only requirement is that semi-supervision is feasible, i.e., that there exists a small amount of paired data in

set $\mathcal{P}_{data}$ with $M_{SSL}$ matched samples from both domains, as described in Section 3.1 and illustrated in Figure 2.

BTS of the $D_X$'s training pool, i.e., BTS of its statistics, is periodically repeated during the whole learning procedure in order to exploit the improved versions of the utilized generator $G_{Y \to X}$ each time. Furthermore, available set of paired, i.e., labeled samples $\mathcal{P}_{data}$ is used alongside the larger unlabeled training sets $X$ and $Y$, from both $\mathcal{X}$ and $\mathcal{Y}$ domains, during the whole training procedure. This means that even after the initial SSL is already accomplished, i.e., after the proposed BTS strategy is invoked for the first time during the training, $\mathcal{P}_{data}$ samples are continued to be used in the SSL objective, Equation (12). In such a way, the initial constraints that are expressing the closeness between the "transformed" and the "true" samples in Figure 2, i.e., expressing the quality of generators $G_{Y \to X}$ and $G_{X \to Y}$ in some training iteration $k$, are also preserved in the future – after the BTS of $D_X$ statistics is periodically performed, and the training pool of $D_X$ is re-expanded again.

In Figure 1, the proposed periodical bootstrapping of the $D_X$'s statistics, which is achieved by $G_{Y \to X}$ GAN that is transferring examples from the full into the scarce domain, is illustrated by an additional third arrow, oriented from $\mathcal{Y}$ towards $\mathcal{X}$ and centrally positioned in the functional diagram of the SSL CycleGAN architecture, in the right part of Figure 1. The same process is also illustrated in Figure 3, where it can be seen that the $X_{BTS}$ denotes the set of novel unlabeled training samples that are added to $D_X$'s training pool. The samples are randomly generated based on the parameters of the generator $G_{Y \to X}$ that have been learned up to that moment.

After performed BTS, the training pool $\{X \cup X_{BTS}\}$ of $D_X$ can be considered as being closer in size to the training pool $Y$ of the discriminator $D_Y$, Figure 3. Thus, despite the unfavourable training conditions, an improved CycleGAN performance is expected to be achieved in comparison to the standard unsupervised training, as well as in comparison to the basic SSL strategy without BTS. It should be mentioned that $X_{BTS}$ is replaced each time with a new set of samples when $G_{Y \to X}$ is periodically invoked for the BTS—randomly generated samples from previous BTS iterations are not accumulated in the training pool of $D_X$.

In summary, after a number of initial iterations, when the discriminator $D_X$, as well as the generator $G_{Y \to X}$ for the scarce domain $\mathcal{X}$, are sufficiently pre-trained by using the combination of adversarial and SSL loss, Equation (12), the described BTS strategy for improving the learning process comes into play. In essence, it results in improving the statistics of $\hat{p}_X(x)$ that is describing the unlabeled data in the training poll $\{X \cup X_{BTS}\}$ of $D_X$. Improved statistics result in better discrimination of $x$, and consequently in the better performance of whole CycleGAN SSL.

Although it has already been discussed, we would like to point out once again that the "unlabeled" refers to training samples of $\{X \cup X_{BTS}\}$ and $\{Y\}$ for which the exact match in the opposite domain is not known exactly and in advance, in contrast to the paired or "labeled" training pools $\{X_p\}$ and $\{Y_p\}$, i.e., set of paired samples $\mathcal{P}_{data}$, which is used in the SSL objective $\mathcal{L}_{SSL}$ in Equations (11) and (12), Figure 2. In that sense, the BTS strategy directly affects only the discriminator term $D_X$ in the cost functions in Equation (12),, i.e., $\mathcal{L}_{adv}(G_{Y \to X}, D_X)$. However, the indirect effect of the increased sample size in the training pool of $D_X$ is successfully propagated to all model elements in Figure 1 during the subsequent training iterations, until the next BTS with a likely better training pool is generated again.

In practice, the proposed BTS strategy consists of a function call to $G_{Y \to X}$ in every $k + K$-th training iteration, for some fixed period $K$. Each BTS results in a fixed amount of novel samples added to $X$, where samples represent, e.g., images in the image translation task. Images in $X_{BTS}$ are generated by the GAN network $G_{Y \to X}^{(k)}$, where $(k)$ denotes the state of parameters in $k$-th training iteration. This is done by using a uniform distribution and randomly choosing a fixed percentage of images in the training set $Y$, which are than transferred to scarce domain by $G_{Y \to X}^{(k)}$. Thus, in every $k + K$-th iteration, the discriminator $D_X$ is trained on a pool $\{X \cup X_{BTS}\}$, augmented in the previously described manner.

### 3.3. Proposed SSL+BTS Training Procedure

In the following lines are formally summarized the main steps of the introduced SSL and BTS strategies, described in Sections 3.1 and 3.2.

Algorithm 2 begins by the order of steps that in the very beginning perform exclusively SSL strategy, during the first $K_0$ out of $N$ iterations, $K_0 \ll N$. Only after the SSL secures that discriminator $D_X$ is not overfitting, the BTS strategy is invoked for the first time in iteration: $K_0 + 1$. Afterwards, the BTS is periodically invoked every $K$-th iterations, and jointly employed with the SSL strategy till the end of the procedure.

---

**Algorithm 2** BTS-SSL CycleGAN training

---

**procedure** CYCLEGAN SSL+BTS (process in the right part of Figure 1)

$N$, number of iterations; $K_0$, number of initial SSL iterations without BTS, $K_0 \ll N$; $K$, period of BTS repetition; $m$, minibatch size; $M_{SSL} = |\mathcal{P}_{data}|$, number of paired examples in $\mathcal{P}_{data} = \{(x_i^p, y_i^p) | i = 1, \ldots, M_{SSL}\} = X_p \times Y_p$; $m_{SSL}$, SSL minibatch size, $m_{SSL} < M_{SSL}$; $q \in (0,1)$, fixed percentage of training data $Y$ that are randomly chosen to produce the bootstrapping samples for the training pool of $D_X$; $\eta > 0$, learning rate; $X \in \mathcal{X}$, $Y \in \mathcal{Y}$, unpaired or unlabeled training sets, while $X_p \in \mathcal{X}$, $Y_p \in \mathcal{Y}$ are paired or labeled training subsets, such that $|\mathcal{X}| \ll |\mathcal{Y}|$, $|X| \ll |Y|$, and $|X_p| = |Y_p|$, but $|X_p| \ll |X|$ ;

Randomly initialize parameters of $D_X, D_Y$, and $G_{X \to Y}, G_{Y \to X}$: $\theta_{D_X}, \theta_{D_Y}, \theta_{G_{X \to Y}}, \theta_{G_{Y \to X}}$

**for** $k = 1$ to $N$ **do**

    Sample minibatch of unpaired training data $\{x_1, \ldots, x_m\} \subset X$, $\{y_1, \ldots, y_m\} \subset Y$

    Sample minibatch of paired training data $\{(x_1^p, y_1^p), \ldots, (x_{m_{SSL}}^p, y_{m_{SSL}}^p)\} \subset X_p \times Y_p$

    $\widehat{\mathcal{L}}_{adv}(G_{X \to Y}, D_Y) = \frac{1}{m} \sum_{i=1}^m \ln D_Y(y_i) + \frac{1}{m} \sum_{i=1}^m \ln \left(1 - D_Y(G_{X \to Y}(x_i))\right)$

    $\widehat{\mathcal{L}}_{adv}(G_{Y \to X}, D_X) = \frac{1}{m} \sum_{i=1}^m \ln D_X(x_i) + \frac{1}{m} \sum_{i=1}^m \ln \left(1 - D_X(G_{Y \to X}(y_i))\right)$

    $\widehat{\mathcal{L}}_{SSL} = \frac{1}{m} \sum_{i=1}^m \left[ \|G_{X \to Y}(x_i) - y_i\|_1 + \|G_{Y \to X}(y_i) - x_i\|_1 \right]$

    $\mathcal{L}_1 = \widehat{\mathcal{L}}_{adv}(G_{Y \to X}, D_X) + \lambda_{cyc} \mathcal{L}_{cyc}(G_{X \to Y}, G_{Y \to X}) + \lambda_{SSL} \widehat{\mathcal{L}}_{SSL}(G_{X \to Y}, G_{Y \to X})$

    $\mathcal{L}_2 = \widehat{\mathcal{L}}_{adv}(G_{X \to Y}, D_Y) + \lambda_{cyc} \mathcal{L}_{cyc}(G_{X \to Y}, G_{Y \to X}) + \lambda_{SSL} \widehat{\mathcal{L}}_{SSL}(G_{X \to Y}, G_{Y \to X})$

    $\theta_{D_X}^{(k+1)} \longleftarrow \theta_{D_X}^{(k)} - \eta \nabla_{\theta_{D_X}} \widehat{\mathcal{L}}_{adv}(G_{Y \to X}, D_X)$

    $\theta_{D_Y}^{(k+1)} \longleftarrow \theta_{D_Y}^{(k)} - \eta \nabla_{\theta_{D_Y}} \widehat{\mathcal{L}}_{adv}(G_{X \to Y}, D_Y)$

    $\theta_{G_{Y \to X}}^{(k+1)} \longleftarrow \theta_{G_{Y \to X}}^{(k)} - \eta \nabla_{\theta_{G_{Y \to X}}} \mathcal{L}_1$

    $\theta_{G_{X \to Y}}^{(k+1)} \longleftarrow \theta_{G_{X \to Y}}^{(k)} - \eta \nabla_{\theta_{G_{X \to Y}}} \mathcal{L}_2$

    **if** $\left(k == K_0 + 1\right) \vee \left((k > K_0) \wedge (((k - K_0) \mod K) == 0)\right)$ **then**

        Bootstrapping the training pool $X_D$ of discriminator $D_X$.

        $X_D \leftarrow \{X \cup X_{BTS}\}$, $X_{BTS} = rand_q\left(G_{Y \to X}^{(k)}(Y)\right)$, where $rand_q(\cdot)$ represents the operator of choosing $\lceil q |Y| \rceil$ samples from the training pool $Y \in \mathcal{Y}$, and transforming them by $G_{Y \to X}^{(k)}$ to scarce domain $\mathcal{X}$, i.e., generating $X_{BTS}$.

    **end if**

  **end for**

**end procedure**

---

In that sense, the adopted abbreviation "SSL+BTS" in Algorithm 2 better reflects the order of appearance of SSL and BTS strategies during the learning process. However, since BTS is periodically repeated, and also represents an extension of the SSL CycleGAN setup, the method is named as BTS-SSL CycleGAN.

Algorithm 2 has all elements of standard CycleGAN training procedure given by Algorithm 1 in Section 2.3. In that sense it can be used in any task where the unsupervised learning approach based on the cycle-consistency loss has already been applied before.

As already explained in Section 3.2, the main characteristic of the presented SSL+BTS procedure is that every $K$ iterations after $k = K_0$ the training pool $X_D$ of discriminator $D_X$ is updated with $\lceil q |Y| \rceil$ synthetic samples generated by $G_{Y \to X}^{(k)}$, i.e., $X_D \leftarrow X \cup X_{BTS}$. However, $X_{BTS}$ is used only for $D_X$, and is not added to unlabeled training set $X$.

Improved performance of $D_X$ indirectly boosts the entire training procedure and leads to better results of the domain transfer in comparison to the baseline learning Algorithm 1, as reported by the results of the conducted experiments in Section 5.

## 4. Elements of the Adopted Experimental Setup

In this section, we provide a description of the elements that were used in order to obtain experimental results presented in Section 5. The experimental setup was designed in such a way to provide extensive comparison between the proposed BST-SSL CycleGAN method and the baseline CycleGAN, as well comparison against one supervised domain transfer method. In order to get better insights into the performance of the proposed SSL+BTS training procedure, experiments included different types of assessments, both quantitative and visual. Therefore, in the following we point out the main characteristics of the individual elements of the adopted experimental setup and provide details of the corresponding training procedures.

### 4.1. Network Architecture and Training Details

For the purpose of experimental comparisons we have implemented the corresponding neural networks based on the best practices from the literature. Since the proposed semi-supervised learning is generally applicable to any variant of the original CycleGAN architecture, we have decided to rely on the referent implementations form the papers were the corresponding methods were originally introduced.

Thus, for the main neural network architecture of the corresponding GANs we have adopted the architecture proposed in [17], where the CycyleGAN model was originally proposed), and which was also utilized in [19]. This implementation of unsupervised domain translation was also the basis for implementation of the SSL+BTS method proposed in this work. In addition, since the original CycyleGAN was introduced in the context of image-to-image translation tasks, the same type of domain transfer tasks was also chosen for the experimental comparisons in this work. One of the advantages of this type of experiments is also that they provide visual insights into the quality of obtained domain transfer results.

The main characteristics of the adopted architecture are that it contains two stride-2 convolutions, as well as several residual blocks and 2 fractionally strided convolutions with stride $\frac{1}{2}$. It uses 6 blocks for $128 \times 128$ and 9 blocks for $256 \times 256$ and higher-resolution type images. It also uses instance normalization as in [17,23,24].

For the discriminator network, we have relied on $70 \times 70$ PatchGAN in order to classify whether the image patches are real or fake, as it was more efficient than the usual full-image network.

When it comes to training details, besides the main algorithmic steps of the performed training procedures, which have already been described in detail in Sections 2.2 and 3, some of the specific settings and values of control parameters that were used in the experiments include the following. In the training procedure, as in [17], instead of negative log likelihood loss figuring in (1), we used more stable $L_2$ loss approach consisting of training $G$ to minimize $E_{z \sim p(z)}[(D(G(z)) - 1)^2]$ as well as training discriminator $D$ to minimize $E_{x \sim p(x)}[(D(x) - 1)^2] + E_{z \sim p(z)}[D(G(z))]$. Furthermore, similarly as in [17], we use the history of generated images (50 of them) in order to calculate the average score. We also keep $\lambda_{cyc} = \lambda_{SSL} = 10$, for all of our experiments. Sizes of minibatches $m$ and $m_{SSL}$ are kept equal and set to $m = m_{SSL} = 50$. Random initialization of the network parameters

was performed by using random samples drawn from normal distribution $\mathcal{N}(0, 0.02)$. All networks were trained using the learning rate of $\eta = 0.0002$ which is kept constant during the first 100 epochs and then linearly decays to zero during the next 100 epochs.

In experiments with all used datasets we have controlled the size of the scarce domain, by changing the percentage $\mathcal{S}_X$ of the domain used: 25%, 50% and 100%, and performed both the unsupervised, as well as the proposed SSL approach (loss function (12)), with and without the proposed bootstrapping of the statistic of the discriminator $D_X$ from the scarce domain (variants SSL+BTS, and SSL). In all experiments, as all datasets used actually contain paired images, we use a fixed 20% of the scarce domain training paired examples for evaluation in the SSL manner. In each experiment, after initial $K_0$ iterations, during the next $k + K$-th training iteration, where $K = 50$, 20% of randomly chosen examples from the target domain is transformed by $G_{Y \to X}$ and added to the pool of the discriminator $D_X$.

### 4.2. Considered Domain Translation Tasks

The particular tasks, as well as datasets that are used in experiments, are the following: Semantic label↔photo task on *CityScapes* dataset [13,25]. The dataset consists of 2975 training images of the size $128 \times 128$, as well as evaluation set for testing; Architectural labels↔photo task [13,26] on CMP *Facade* dataset, containing 400 training images; as well as Map↔aerial photo task on *Google Maps* dataset [13], containing 1096 training images of the size $256 \times 256$.

All experiments were performed in an imbalanced domain scenario, where we keep the "left" domain (original images domain in all experiments) scarce, i.e., we simulate that particular situation by using only a certain percentage of the original left domain.

### 4.3. Considered Baseline Domain Translation Methods

We evaluate the proposed approach in comparison to well-established baseline image-to-image translation methods, where we choose the original CycleGAN method proposed in [17], as well as pix2pix image translation method proposed in [13].

### 4.4. Utilized Measures for Results Comparison

Evaluating the quality of synthesized images is an open and difficult problem. We use some classical measures such as Peak Signal-to-Noise Ratio (PSNR) as well as more advanced Structural Similarity Index Measure (SSIM), which is a perception-based model that considers image degradation as perceived change in structural information. It also incorporates important perceptual phenomena, including both luminance masking and contrast masking terms. The SSIM measure is much more appropriate for measuring image degradation than PSNR.

The PSNR is evaluated as $PSNR = 20log\left(MAX_I / \sqrt{MSE}\right)$, where $MAX_I$ is the maximum possible pixel value of the ground truth images, while $MSE$ is the squared Euclidean norm between the generated and ground truth images. The SSIM measure between images generated by the considered GAN algorithms and ground truth images is calculated on various windows $x, y$ of an image, by using the following formula:

$$SSIM(x, y) = \frac{(2\mu_x\mu_y + c_1)(2\sigma_{xy} + c_2)}{(\mu_x^2 + \mu_y^2 + c_1)(\sigma_x^2 + \sigma_y^2 + c_2)} \tag{13}$$

where $\mu_x$ and $\mu_y$ are average values of $x$ and $y$, while $\sigma_x^2$ and $\sigma_y^2$ are variances and $c_1$ and $c_2$ are constants, set as reported in [27].

In addition to the mentioned classical image quality measures, we have also used some more specific measures, such as measures based on human judgments, which we call Perceptual Realistic Measure (PRM), as well as the FCN score measure proposed in [28], which is based on pre-trained semantic classifiers and measures how discriminative the generated output is. We then score the synthesized photos using the classification accuracy against the labels these photos were synthesized from.

There are various measures of FCN accuracy used in [28]: pixel accuracy evaluated as $\sum_i n_{ii} / \sum_i t_i$, mean accuracy evaluated as $(1/n_{cl}) \sum_i (\sum_i n_{ii}/t_i)$ and mean region intersection over union accuracy evaluated as $(1/n_{cl}) \sum_i n_{ii} / \left( t_i + \sum_j n_{ji} - n_{ii} \right)$. Term $n_{ij}$ is the number of pixels of class $i$ predicted to belong to class $j$, $n_{cl}$ is the number of classes and $t_i = \sum_j n_{ij}$ is the total number of pixels belonging to the class $i$.

Concerning the PRM measure, we have asked student participants (10 of them) at the Faculty of Technical Sciences, University of Novi Sad, to evaluate the perceptual quality of images generated by the proposed method in comparison to baseline methods. We note that for us it was much more accessible to organize our own pool of human participants, than to use some internet service such as AMT perceptual studies. In the Map↔ aerial photo experiments conducted on *Google Maps*, we follow the procedure reported in [17].

The participants were shown pairs of images, each pair containing one real image (i.e., the ground truth) and one synthesized, generated by the proposed as well as baseline algorithms. Next, they were asked to choose an image they thought it was real. The first 5 trials of each session were conducted for practice and a feedback was given whether the participant's response was correct or incorrect. The remaining 15 trials were used to assess the rate at which each algorithm fooled the participants.

In addition, the relative improvement of the proposed method against the CycleGAN baseline, denoted by $\delta$, was computed for each of the described measures separately.

## 5. Experimental Results

In this section, we present experiments on several real image datasets in the task of image-to-image translation, showing the efficiency of the proposed domain transformation method that we call BTS-SSL CycleGAN.

### 5.1. Quantitative Evaluation of the Conducted Experiments

Results of the conducted experiments are summarized in Tables 1 and 2. The relative improvement brought by the proposed method in Table 1 is expressed by average $\delta_{\text{PSNR}}$ and $\delta_{\text{SSIM}}$, while SSL and BTS denote results of the methods corresponding to individual elements of the proposed SSL-BTS learning strategy.

**Table 1.** Experimental comparison between proposed semi-supervised (SSL+BTS) and the unsupervised CycleGAN, under different scenarios: varying sample size $\mathcal{S}_X$ of scarce domain $\mathcal{X}$, as well as when applied to different tasks/datasets—*CityScapes, Facade dataset, Google Maps*.

| | $\mathcal{S}_X$ | pix2pix | | CycleGAN | | SSL | | BTS | | SSL+BTS | | $\delta_{\text{PSNR}}$ | $\delta_{\text{SSIM}}$ |
|---|---|---|---|---|---|---|---|---|---|---|---|---|---|
| | [%] | PSNR | SSIM | PSNR | SSIM | PSNR | SSIM | PSNR | SSIM | PSNR | SSIM | [%] | [%] |
| *CityScapes* | 25 | **19.98** | 0.60 | 17.20 | 0.56 | 18.30 | 0.59 | 17.30 | 0.58 | 18.77 | 0.61 | 9.1 | 8.9 |
| | 50 | **20.45** | 0.64 | 17.00 | 0.55 | 18.95 | 0.61 | 17.18 | 0.59 | 19.04 | 0.64 | 12.00 | 16.4 |
| | 100 | 19.51 | 0.59 | 17.12 | 0.54 | 20.03 | 0.58 | 17.75 | 0.63 | **20.47** | 0.65 | 19.6 | 20.4 |
| *Facade dataset* | 25 | **13.78** | 0.35 | 10.93 | 0.25 | 11.78 | 0.31 | 10.81 | 0.27 | 11.83 | 0.32 | 8.23 | 28.0 |
| | 50 | **14.24** | 0.40 | 11.00 | 0.25 | 13.22 | 0.37 | 11.92 | 0.28 | 13.75 | 0.40 | 25.0 | 60.0 |
| | 100 | **14.25** | 0.42 | 10.98 | 0.27 | 12.88 | 0.33 | 11.52 | 0.35 | 13.21 | 0.41 | 20.3 | 51.8 |
| *Google Maps* | 25 | 30.35 | 0.67 | 30.47 | 0.71 | 30.62 | 0.73 | 30.55 | 0.75 | **31.20** | 0.77 | 2.4 | 8.4 |
| | 50 | 30.55 | 0.68 | 29.78 | 0.72 | 30.68 | 0.75 | 29.81 | 0.76 | **30.88** | 0.79 | 3.7 | 9.7 |
| | 100 | 30.01 | 0.69 | 30.24 | 0.73 | 30.92 | 0.75 | 30.27 | 0.77 | **31.23** | 0.81 | 3.3 | 11.0 |

Note: maximum for each $\mathcal{S}_X$ is typeset in bold (PSNR) and blue bold with blue highlight (SSIM), while the largest $\delta$ over all $\mathcal{S}_X$ is shaded in green.

BTS training includes only bootstrapping of the discriminator $D_X$, without paired samples, while SSL involves using paired samples, but without bootstrapping. As an example of the performance of supervised methods on the same tasks, results of pix2pix method are also presented.

In Table 1 the results of the PSNR, as well as SSIM measure between images generated by the considered GAN algorithms (baseline pix2pix, CycleGAN as well as the proposed BTS-SSL CycleGAN, in an image-to-image translation task, from scarce to target domain) and ground truth images, are presented, for all the dataset used.

From Table 1 it can be seen that the proposed BTS-SSL CycleGAN algorithm obtains better results on average, by both PSNR and SSIM measures, in all of the experiments, in comparison to the CycleGAN method, especially on *CityScapes* and CMP *Facade* datasets. It can also be seen that both SSL and BTS components of the proposed algorithm contribute significantly to the obtained results.

In Table 2, the PRM results for the *Google Maps* dataset and the Photo$\rightarrow$ Map task, for the proposed method in comparison to the baseline methods, are given in form of the accuracy of detecting generated, i.e., false images by the participants, and expressed in %.

**Table 2.** Results of the second type of experiments. Performance of the proposed SSL+BTS was compared against unsupervised CycleGAN, and supervised pix2pix by using Perceptual Realistic Measure (PRM) for the 'Photo $\rightarrow$ Map' domain translation task, and by assessing image segmentation quality in the case of 'Photo $\rightarrow$ Semantic label' task. Relative improvement of SSL+BTS over CycleGAN is expressed by $\delta$, while the respective datasets are listed in the table.

| Task/Dataset | Measure | $\mathcal{S}_X$ | pix2pix | CycleGAN | SSL | BTS | SSL+BTS | $\delta$ |
|---|---|---|---|---|---|---|---|---|
| Photo $\rightarrow$ Map *Google Maps* | PRM | 25 | **18.8** | 17.5 | 18.0 | 18.1 | 18.4 | 5.1 |
| | | 50 | **19.8** | 18.7 | 19.3 | 19.2 | 19.5 | 4.3 |
| | | 100 | **21.7** | 20.4 | 20.8 | 20.7 | 21.1 | 3.4 |
| Photo $\rightarrow$ Semantic label *CityScapes* | FCN per-pixel acc. | 25 | **0.57** | 0.43 | 0.45 | 0.46 | 0.50 | 16.3 |
| | | 50 | **0.63** | 0.49 | 0.52 | 0.51 | 0.54 | 10.2 |
| | | 100 | **0.71** | 0.52 | 0.56 | 0.55 | 0.58 | 11.5 |
| | FCN mean acc. | 25 | 0.16 | 0.13 | 0.15 | 0.16 | **0.18** | 38.5 |
| | | 50 | **0.20** | 0.15 | 0.17 | 0.18 | 0.19 | 26.7 |
| | | 100 | **0.25** | 0.17 | 0.19 | 0.20 | 0.22 | 29.4 |
| | FCN mean IoU | 25 | **0.13** | 0.09 | 0.10 | 0.10 | 0.12 | 33.3 |
| | | 50 | **0.16** | 0.10 | 0.11 | 0.12 | 0.13 | 30.0 |
| | | 100 | **0.18** | 0.11 | 0.13 | 0.13 | 0.15 | 36.4 |

Note: all values are expressed in [%]; higher PRM is better (% of gen. images misinterpreted as real); maximum for each $\mathcal{S}_X$ is typeset in bold and the largest $\delta$ over all $\mathcal{S}_X$ is shaded in green.

It can be seen that, similarly to other experiments, the proposed BTS-SSL CycleGAN outperforms the baseline CycleGAN, but as expected, does not outperform the fully supervised pix2pix method. Furthermore, both BTS as well as SSL components of the method contribute to the final result.

Results of FCN accuracy measure for the proposed BTS-SSL CycleGAN in comparison to baseline methods are provided in Table 2 for the *CityScapes* dataset and the Semantic label$\leftrightarrow$photo task, are presented. It could be seen that the proposed BTS-SSL CycleGAN obtains better results in comparison to the CycleGAN, but it is outperformed by the fully supervised pix2pix, as expected. It could also be seen that both BTS as well as SSL components of the algorithm contribute to the final result.

*5.2. Visual Comparison between Analyzed Methods*

As an example of the conducted visual comparisons, in the following are presented several figures that illustrate visual performance of the proposed BTS-SSL CycleGAN method against considered baselines.

In Figures 4–6, visual examples are given for the proposed BTS-SSL CycleGAN vs. baseline algorithm comparisons, for 100% of the scarce domain data used: Real A (scarce domain) and Real B (target domain) correspond to image pair examples. Examples are

shown for *Google Maps*, *CityScapes* and *Facade* datasets. It can be seen that in those examples, the proposed BTS-SSL CycleGAN obtains visually more accurate results than the baseline methods.

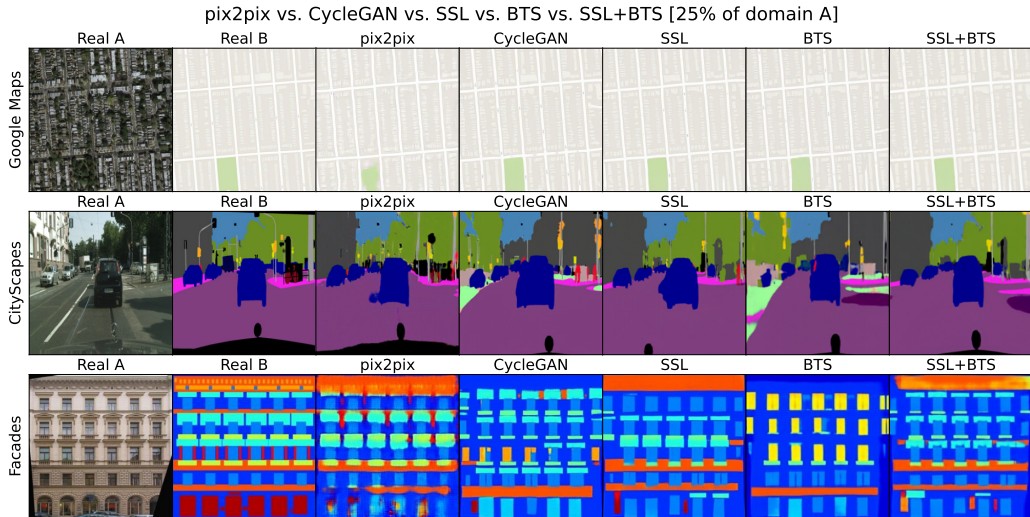

**Figure 4.** Visual examples of the proposed vs. baseline algorithm comparisons for 25% of the scarce domain data used: Real A (scarce domain) and Real B (target domain) correspond to image pair examples. Examples are given for *Google Maps*, *CityScapes* and *Facade* datasets. SSL denotes that only SSL on the control part of pared images is included in the training of CycleGAN, BTS denotes that only bootstrapping of the statistic of the A discriminator pool is included in the training, while SSL+BTS denotes that both SSL and BTS are included.

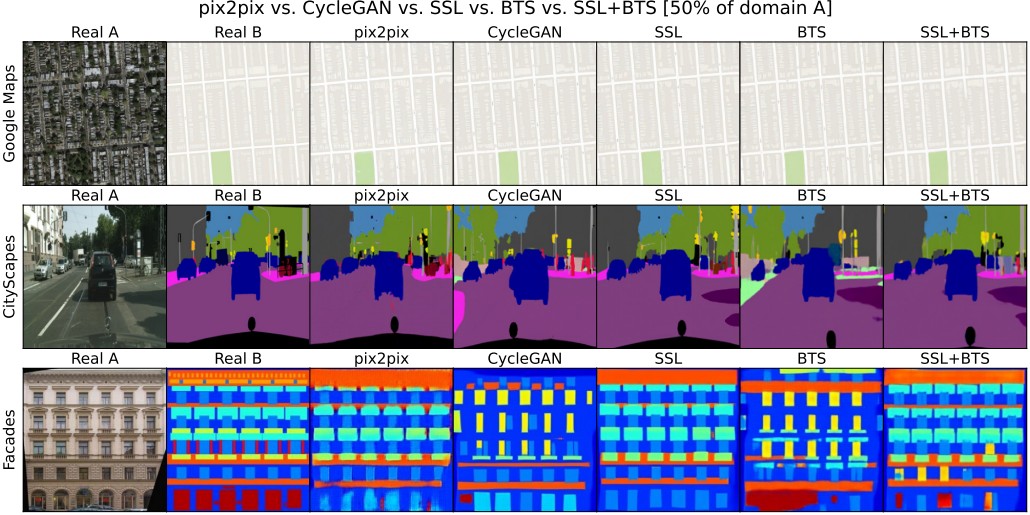

**Figure 5.** Visual examples of the proposed vs. baseline algorithm comparisons for 50% of the scarce domain data used: Real A (scarce domain) and Real B (target domain) correspond to image pair examples. Examples are given for *Google Maps*, *CityScapes* and *Facade* datasets. SSL denotes that only SSL on the control part of pared images is included in the training of CycleGAN, BTS denotes that only bootstrapping of the statistic of the A discriminator pool is included in the training, while SSL+BTS denotes that both SSL and BTS are included.

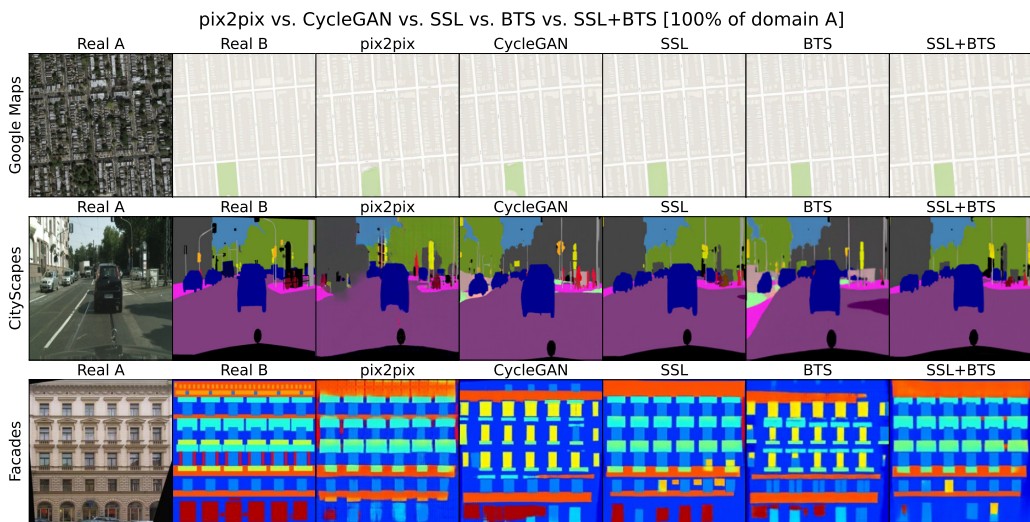

**Figure 6.** Visual examples of the proposed vs. baseline algorithm comparisons for 100% of the scarce domain data used: Real A (scarce domain) and Real B (target domain) correspond to image pair examples. Examples are given for *Google Maps*, *CityScapes* and *Facade* datasets. SSL denotes that only SSL on the control part of pared images is included in the training of CycleGAN, BTS denotes that only bootstrapping of the statistic of the A discriminator pool is included in the training, while SSL+BTS denotes that both SSL and BTS are included.

## 6. Conclusions

In this work, in order to deal with imbalanced domains problem in the context of domain translation or domain transfer tasks, we have proposed a novel solution for overcoming the difficulties of the unsupervised learning process in such case. The proposed method combines the unsupervised CycleGAN architecture with SSL learning strategy that is additionally improved through an internal bootstrapping procedure (BTS). Thus, the BTS-SSL CycleGAN semi-supervised domain translation model was formed. Based on experimental results, it has exhibited potential to be a method of choice for the improvement of any CycleGAN-based domain translation task, especially in the considered imbalanced domains scenario. The proposed SSL strategy was used during the initial training stages to prevent discriminator related to the scarce domain from overfitting, while the employed BTS strategy for bootstrapping the statistics of the training pool of the discriminator was used to improve the learning performance in the unfavourable scenario of imbalanced domains. BTS was performed by inserting the example data generated by the corresponding internal generator, i.e., mapping from the fully observable into the scarce domain. We have manage to obtain significantly better results in comparison to the original CycleGAN method, and comparable with the fully supervised pix2pix method, on several image domain transfer datasets. In further work, we will focus on other more intelligent ways to bootstrap the statistics of the discriminator of the scarce domain, by inserting into its pool only a certain amount of transferred image examples that "fit" the initial statistics of the mentioned pool sufficiently well.

**Author Contributions:** Conceptualization, M.J., L.K., B.P. and B.B.; Data curation, B.P.; Formal analysis, L.K.; Funding acquisition, L.K. and B.P.; Investigation, M.J. and B.B.; Methodology, M.J., L.K. and B.P.; Project administration, L.K.; Software, B.P.; Supervision, L.K. and B.P.; Visualization L.K,; Writing—original draft, M.J. and L.K.; Writing—review & editing, B.B., L.K. and B.P. All authors have read and agreed to the published version of the manuscript.

**Funding:** This research was funded by the "Science Fund of the Republic of Serbia", through the project grant agreement No. 6524560: "Speaker/Style Adaptation for Digital Voice Assistants Based on Image Processing Methods (AI-S-ADAPT); and by the "Serbian Ministry of Education, Science and

Technological Development" through the research project No. 451-03-68/2020-14/200156: "Innovative Scientific and Artistic Research from the Faculty of Technical Sciences Activity Domain".

**Institutional Review Board Statement:** Not applicable.

**Informed Consent Statement:** Not applicable.

**Data Availability Statement:** All data sources are publicly available and mentioned in the references.

**Acknowledgments:** This research was supported by the "Science Fund of the Republic of Serbia", through the project grant agreement No. 6524560: "Speaker/Style Adaptation for Digital Voice Assistants Based on Image Processing Methods (AI-S-ADAPT), and by the "Serbian Ministry of Education, Science and Technological Development" through the research project No. 451-03-68/2020-14/200156: "Innovative Scientific and Artistic Research from the Faculty of Technical Sciences Activity Domain". The authors would also like to acknowledge support by the EU's H2020 research and innovation programme under the project grant agreement No. 872614: "Selfsustained Cross Border Customized Cyberphysical System Experiments for Capacity Building Among European Stakeholders" (Smart4All).

**Conflicts of Interest:** The authors declare that there is no conflict of interest.

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
