# Peer review of "Bootstrapped SSL CycleGAN for Asymmetric Domain Transfer"

_applsci, doi:10.3390/app12073411_

Round 1

Reviewer 1 Report

This paper presents an improved cyclegan method, which is very meaningful. It is a good improvement, and the results of the article also confirm the effectiveness of the improvement.
However, further modifications are needed before publication:
1)This paper only compares the comparison results between the improved method and the  CycleGAN method, but does not give the comparison results between other methods and the proposed method. Is the improvement of this paper also suitable for the improvement of other methods?
2)There are some grammatical errors and format errors in the article, which need to be carefully checked by the author.
3)It is suggested that the network architecture and training details should be described in detail, rather than providing only two algorithm processes.

Author Response

Dear Reviewer,

Responses to the Reviewers' comments are provided in the attachment  of this letter.

Authors

Reviewer 2 Report

This article describes a fascinating approach to an emerging process of analysis. Though highly technical (and therefore very useful) it is also clearly indicating advances in image analysis and comparison. 

Author Response

(The authors gave the same response as above.)

Reviewer 3 Report

In the paper authors are presenting a Bootstrapped SSL CycleGAN architecture to reduce the standard imbalance domain problem in the domain transfer task. This is relevant problem and it seems that proposed approach yields competitive results.

However, structure of the paper and presentation style should be significantly improved. Although two algorithms are included in the paper, in my opinion, complete description of the procedure is too narrative and it should be presented in more structured (step by step) way.

As an example of presentation style that needs to be improved, in line 103, authors write "Thus, by applying the previous during..."  which is quite vague.

Reasons and explanations should be more precise.

In line 21 references are for style transfer written as [12]-[7]. What does it mean?

Perhaps the most important objection is to the results section. Presentation is under acceptable level. There are too many small tables (all results could be presented with one or two larger tables). Tables are also hard to read, with redundancies (% sign in column titles and below), lacking descriptions in table and column titles. It is not enough to write description under the table, table should be easy to understand by itself. Also, suggestion is to accentuate best results. Again, text describing the results is narrative and could be better structured.

Author Response

(The authors gave the same response as above.)

Round 2

Reviewer 1 Report

The author has responded well to relevant questions and made corresponding modifications. I think it can meet the requirements of publishing.

Reviewer 3 Report

All comments by the reviewer were properly answered and modifications were done. I don't have further comments.